# OpenReview forum: "ProxyPrompt: Securing System Prompts against Prompt Extraction Attacks"
_ICLR.cc/2026/Conference — ICLR 2026 Conference Withdrawn Submission_

### Official Review · Reviewer_aUe5 · 2025-10-30

**Soundness:** 3
**Presentation:** 3
**Contribution:** 3
**Rating:** 6
**Confidence:** 4

**Summary:**

This paper aims to propose a new defense against prompt extraction attacks which can ensure the utility and the extraction prevention simultaneously. Specifically, they first design a new soft token learning strategy under a combined objective function. Then, they propose several new metrics to measure the proportion of prompt leakage. Under the experiments on different datasets, they show that their method can consistently outperform existing methods and demonstrates a better tradeoff between utility and extraction prevention.

**Strengths:**

+ This paper proposes a novel strategy for defending against prompt extraction attacks.
+ The proposed method consistently outperforms existing methods.
+ The paper clearly illustrates their methodology and the detailed experimental details in their evaluation.

**Weaknesses:**

+ The proposed method can only be used under a threat model that the defender can obtain the parameter information of LLMs. This assumption is not always fulfilled. For instance, sometimes the LLM provider might not consider the defense, while the developer of an agent or an application would like to seek some strategy to protect their prompts. Under such a situation, this method is not applicable.
+ The proposed method requires significant training resources to enable the input embedding optimization. Can the authors specify which computation hardware they used for each backbone model?

**Questions:**

Please refer to the weaknesses.

---

> ### Author Response · Authors · 2025-11-20
>
> We thank the reviewer for recognizing our work as a novel defense strategy against prompt extraction attacks with strong experimental results that consistently outperform existing methods. We now address the remaining points below.
>
>
> > W1: Discussion for closed-source LLM.
>
> We acknowledge the concern regarding scenarios where the LLM provider does not support the defense. We hope ProxyPrompt could highlight the value of open-source models, where developers have full flexibility to apply such protection. As noted in Line 78, we also encourage future API-level support from closed-source providers to remain competitive with open-source ones. For instance, they could offer a prompt optimization API without exposing model weights, similar to OpenAI’s fine-tuning API [1]. In such cases, ProxyPrompt would also be applicable.
>
>
> [1] OpenAI. Gpt-3.5 turbo fine-tuning and api updates. https://openai.com/blog/gpt-3-5-turbo-fine-tuning-and-api-updates, 2023. Accessed: 2025-1-18.
>
>
> > W2: ...Can the authors specify which computation hardware they used for each backbone model?
>
> All experiments were conducted on a single NVIDIA H200 GPU with 141 GB of memory and an Intel Xeon CP, as reported in Appendix G. The optimization time can be further reduced by parallelizing across multiple GPUs.

---

> > ### Comment · Reviewer_aUe5 · 2025-11-26
> >
> > Dear authors,
> >
> > Thank you for your response, and please accept my apologies for the late comments.
> >
> > After reading your reply, I suggest that you add W1 as a limitation in the main text, since you also acknowledge it. From a practical standpoint, most agents are built on top of LLM APIs without further training, so this is an important point: most developers cannot modify the backbone model to protect their prompts from disclosure. My W2 raises the same concern.
> >
> > Overall, I remain positive about this submission.
> >
> > Thanks.

---

### Official Review · Reviewer_CQip · 2025-10-30

**Soundness:** 2
**Presentation:** 2
**Contribution:** 2
**Rating:** 2
**Confidence:** 4

**Summary:**

This paper obfuscates the system prompt to prevent prompt injection attack.

**Strengths:**

+ The topic is important for LLM security.

**Weaknesses:**

- Limited novelty. While the application is defensive, the core technique of optimizing in the continuous embedding space to steer model output is conceptually similar to well-established work. Methods like GCG, Prompt Obfuscation already explore this approach.

- Insufficient baseline. This is a critical flaw in the evaluation. The chosen baselines (FILTER, FAKE, DIRECT) are all lightweight, prompt-engineering techniques that require no model optimization and add negligible computational overhead. ProxyPrompt, in contrast, requires a significant optimization process that consumes considerable computational resources (e.g., 6 hours on an H200 GPU for one GSM8K prompt). A more equitable and insightful comparison would involve benchmarking against other defense methods that also require a training or optimization phase. For instance, finetuning-based method like SecAlign or StruQ. Also, you need to compare with strong prompting-based approach, such as top DSR prompts used in TensorTrust, instead of a simple defensive prompt.

- Hard to scale to long system prompt. The experiments raise serious doubts about whether it can effectively protect the long and complex system prompts that are increasingly common and valuable in real-world applications. The paper use 16 tokens for GSM8k and claims that the full 8-shot system prompt is only 834 tokens long (I think it does not contain COT .
It is much shorter than real-world prompts, such as GPT-5 system prompt leaked.
This limitation is particularly concerning for agentic workflows (e.g., OpenHands, SWE-Agent), where system prompts can be thousands of tokens long to define tool interactions and multi-step problem-solving plans. These long prompts represent the most valuable intellectual property and are thus the most critical to protect. The paper provides no evidence that ProxyPrompt can scale to this level of complexity, which is arguably where such a defense is needed most. The information bottleneck imposed by a short proxy seems fundamentally limiting.

**Questions:**

Can you show that ProxyPrompt be able to extend to much longer system prompt while preserving the utility?

---

> ### Author Response · Authors · 2025-11-20
>
> We thank the reviewer for acknowledging the importance of this topic for LLM security and for raising the concerns. We address these concerns below.
>
> > W1: Novelty discussion.
>
> We have closely compared these concurrent work in Related Works (Line 106) and Ablation Study（Line 409). Specifically, Prompt Obfuscation (built on GCG) optimizes prompts with a single objective, enforceing consistent model outputs without constraining how the optimized prompt itself deviates from the original. As a result, their soft prompt may still encode semantically similar information and remain recoverable,
> leading to substantially weaker protection (only 81.1% system prompts protected, compared to 94.7% with ProxyPrompt).
> In contrast, our joint objective explicitly steers the proxy prompt toward a defender-specified target, ensuring both semantic divergence and controlled obfuscation. Consequently, ProxyPrompt not only achieves stronger protection but also lets defenders control the appearance of proxy prompts.
>
>
>
> > W2: Baseline comparison.
>
>
> Methods like SecAlign are complementary to ours: finetuning the model can reduce the chance that the model follows attack instructions, while ProxyPrompt further ensures that even if an attack succeeds, the extracted content is obfuscated and unusable. We also note that all victim models used in our experiments are instruction-tuned versions, already aligned for safety; ProxyPrompt therefore builds on these aligned models to provide an additional layer of defense specifically against prompt extraction.
>
> Regarding prompting-based approaches, we follow established baselines from prior work. The defensive prompts in TensorTrust are manually crafted for a specific “bank-hacking” task, making them unable to be adapted to other tasks in our evaluation. We are glad to include any defensive prompts the reviewer has in mind that can be appended after all our system prompts.
>
>
> > W3 & Q1: Scale to long system prompt.
>
>
> We clarify that the full 8-shot GSM8K system prompt does include full CoT reasoning examples, as shown in Figure 14. For complex multi-step reasoning workflows, we provide a case study on ALFWorld (Appendix M, Figure 29), where solving each task involves more than 50 steps of planning, sub-goal tracking, and exploration, demonstrating ProxyPrompt’s ability to handle long and structured prompts. As discussed in Line 454, ProxyPrompt can be concatenated with non-sensitive prompts to introduce new functionalities without re-optimization. In practice, protecting an entire prompt is often unnecessary, since non-sensitive instructions (e.g., “You are ChatGPT, a large language model trained by OpenAI.”) pose no security risk.
>
> With that being said, we additionally evaluate ProxyPrompt on the [very long Claude system prompt](https://docs.anthropic.com/en/release-notes/system-prompts#claude-3-7-sonnet). We use Llama-3.1-8B-Instruct as the victim model and apply the same setup used in the Roles task. ProxyPrompt achieves strong results in this setting: Utility-Ratio = 0.95, Most-Similar = 0.18, and both Semantic-Match and Approx-Match = 0, indicating effective protection even for lengthy, instruction-rich prompts.

---

> ### Author Response · Authors · 2025-11-30
> **Additional Results for W2: Baseline Comparison**
>
> While we have compared with establised prompt extraction defense methods, we additionally evaluate Llama [Prompt Guard 2](https://huggingface.co/meta-llama/Llama-Prompt-Guard-2-86M) as a new baseline, a classifier-based guard that blocks queries predicted as malicious. Prompt Guard is trained to detect both prompt-injection and jailbreak attacks using **a large corpus of known vulnerabilities**, whereas ProxyPrompt is optimized **only with a single trivial attack query**. As shown below, Prompt Guard fails to protect system prompts: its Approx-Match (AM) score remains 1.0 across all tasks, indicating all prompt are extracted. In contrast, ProxyPrompt achieves 0.00 Approx-Match for all system prompts and the lowest SM (Semantic-Match) score. We will update the paper to include the complete result. We hope this would address the reviewer's concern and update their score.
>
> | model | defense        | GSM8K                         | Roles                            | Cola                           | SST2                            | QNLI                           |
> |-------|----------------|-------------------------------|----------------------------------|--------------------------------|----------------------------------|--------------------------------|
> | L8B   | Prompt Guard 2   | UR=1.00; AM=1.00; SM=1.00; MS=0.96 | UR=1.00; AM=1.00; SM=1.00; MS=1.0  | UR=1.00; AM=1.00; SM=1.00; MS=0.99     | UR=1.00; AM=1.00; SM=0.95; MS=0.97 | UR=1.00; AM=1.00; SM=0.90; MS=0.99 |
> | L8B   | Ours       | UR=0.99; AM=0.00; SM=0.00; MS=0.18 | UR=1.00; AM=0.00; SM=0.00; MS=0.31 | UR=1.01; AM=0.00; SM=0.05; MS=0.40 | UR=1.00; AM=0.00; SM=0.10; MS=0.53 | UR=0.94; AM=0.00; SM=0.05; MS=0.38 |

---

### Official Review · Reviewer_nKxP · 2025-10-31

**Soundness:** 2
**Presentation:** 3
**Contribution:** 2
**Rating:** 4
**Confidence:** 3

**Summary:**

This paper introduces ProxyPrompt, a novel defense mechanism against prompt extraction attacks on large language models (LLMs). Instead of trying to prevent prompt leakage through filtering or instruction-based methods, ProxyPrompt replaces the original system prompt with an optimized proxy in embedding space that maintains functionality for legitimate users but becomes semantically unusable when extracted by attackers. The authors evaluate their approach across 264 configurations spanning multiple models (Phi-3.5-mini, Llama-3.1-8B, Llama-3.1-70B) and tasks (GSM8K, Roles, CoLA, SST-2, QNLI), demonstrating 94.70% protection against extraction attacks—significantly outperforming existing methods. The paper also introduces semantic-level metrics for evaluating prompt leakage and validates the approach on real-world applications including HuggingChat's Image Generator and ALFWorld multi-step reasoning environments.

**Strengths:**

- Novel and impactful approach: The core idea of replacing system prompts with embedding-space-optimized proxies represents a fundamentally different defense strategy compared to existing prompt-based or filtering-based methods, addressing key limitations of prior work.
- Comprehensive evaluation: The extensive testing across diverse models, tasks, and configurations (264 total) provides strong empirical evidence for the approach's effectiveness. The inclusion of real-world case studies (HuggingChat, ALFWorld) enhances practical relevance.
- Strong empirical results: The paper demonstrates clear superiority over baselines (Filter, Fake, Direct) across all metrics, with 94.70% protection rate versus 42.80% for the next-best method.
- Practical considerations: The paper addresses important deployment concerns such as combining proxy prompts with non-sensitive instructions and shows how the method scales for different query set sizes.

**Weaknesses:**

1. Limited attack scenario coverage: The optimization relies on a single trivial attack query (Q'), which may not fully represent sophisticated adaptive attacks. While the paper tests a 10-round adaptive attack on Image Generator, more comprehensive testing against multi-stage, context-aware attacks would strengthen the claims.
2. Computational cost concerns: The optimization process can take up to 24 hours for complex tasks like ALFWorld, which may be prohibitive for large-scale deployment. While parallelization is mentioned, the paper doesn't quantify the practical implications for systems with hundreds of prompts.
3. Insufficient discussion of sensitive high-level intent: The paper notes that classification tasks sometimes leak high-level intent but dismisses this as non-sensitive. However, in many real-world scenarios (e.g., medical diagnosis, content moderation), high-level classification rules are precisely the confidential information that needs protection.

**Questions:**

1. Attack scenario coverage: The paper acknowledges that the current results represent a "lower bound" on performance since the optimization uses a trivial attack query (Q'). Could you provide more details on how the method would perform against more sophisticated adaptive attacks? Specifically, what would be the impact if attackers used multiple rounds of context-aware queries that gradually build a complete picture of the system prompt?
2. Computational cost and scalability: The optimization time for ALFWorld (24 hours) is significant. How would this scale for a service with thousands of system prompts? Are there techniques to reduce this cost (e.g., transfer learning between similar prompts, approximate optimization methods), and what would be the trade-offs in protection effectiveness?
3. Sensitive high-level intent: In classification tasks, the paper states that "high-level intent is often not confidential, while protecting detailed behavior is more critical." However, consider a medical diagnosis system where the high-level classification rules (e.g., "classify as high-risk if age > 65 and symptoms X, Y, Z") are precisely the sensitive information that needs protection. How would ProxyPrompt handle such cases, and could it be adapted to protect these high-level rules?
4. Ethical implications: The ethics statement mentions potential misuse but doesn't elaborate. Could you discuss specific scenarios where ProxyPrompt could be used maliciously (e.g., to hide harmful instructions from oversight), and what safeguards or detection mechanisms might be implemented to prevent such misuse?
5. Comparison with hierarchical instruction schemes: The paper mentions hierarchical instruction schemes (Hines et al., 2024) as complementary to ProxyPrompt but doesn't test them together. How would combining ProxyPrompt with these hierarchical schemes affect overall security and utility, and would this be a recommended best practice for deployment?

---

> ### Author Response · Authors · 2025-11-20
>
> We thank the reviewer for for acknowledging the novelty, impact, and robustness of our approach across models and tasks, as well as the positive feedback on our case studies and practical scalability considerations. We address the remaining points below.
>
> > Q1: Attack scenario coverage
>
> The effectiveness of ProxyPrompt primarily stems from the decoding gap and the joint objective that explicitly steers the appearance of the proxy prompt toward a defender-specified target, rather than from modeling sophisticated adaptive attacks during optimization. Thus, we intentionally use only a single trivial attack query instead of more complex ones in the optimization.
> Our evaluation also includes the adaptive, multi-round attacks mentioned by the reviewer, as detailed in Appendix N. An attacker (GPT-4.1-mini) is informed about the defense and issues multiple innocuous-looking queries to progressively infer the system prompt. The final extraction scores are SM = 0, AM = 0, and MS = 0.49, showing that ProxyPrompt prevents prompt leakage even against adaptive, multi-stage attackers. We also note that even the single-round attacks in the main experiments already cover multiple categories of attack scenarios established in recent works.
>
>
> > Q2: Computational cost and scalability
>
> We acknowledge the computational cost concerns. We think the computational cost is reasonable for protecting high-value prompts. For thousands of system prompts, developers can apply ProxyPrompt only to sensitive prompts and parallelize optimization across GPUs. As discussed in Line 454 (“Adding non-sensitive instructions”), protecting a prompt entirely is often unnecessary because non-sensitive instructions pose no security risk, e.g., “You are ChatGPT, a large language model trained by OpenAI.” We show that proxy prompts remain valid even when non-sensitive instructions are later added, eliminating the need for re-optimization.
>
>
> > W3: Insufficient discussion of sensitive high-level intent.
>
> We clarify that we use "high-level intent" to refer to generic task descriptions such as “classify whether a patient is high-risk.” In contrast, detailed rules like "classify as high-risk if age > 65 and symptoms X, Y, Z" contain sensitive logic that ProxyPrompt can protect. For example, in the Image Generator case (Fig. 1), the system’s rule for generating a Phony Phone image is protected, though the attacker could still infer the intent, "synthesize images for users", which we consider non-sensitive. We will include this discussion in the revised version to improve clarity.
>
>
>
> > Q4: Ethical implications.
>
>
> We acknowledge that our method could be misused if applied irresponsibly. ProxyPrompt might be abused to conceal harmful or biased system prompts, such as instructions that embed gender or racial bias, from oversight. To mitigate misuse, ProxyPrompt should be deployed in transparent, auditable environments where the original system prompts remain accessible to authorized reviewers or compliance tools.
>
> > Q5: Comparison with hierarchical instruction schemes.
>
> We clarify that hierarchical instruction schemes are already incorporated in all our experiments, as noted in Line 126: “All methods in our experiments are evaluated with specialized delimiters (Hines et al., 2024) in the chat template to separate system and user inputs.”

---

> > ### Comment · Reviewer_nKxP · 2025-11-24
> >
> > I appreciate the clarifications regarding attack robustness, computational scalability, the distinction between high-level intent and sensitive logic, ethical considerations, and the integration with hierarchical instruction schemes. I will take your rebuttal into account when forming my final recommendation.

---

> > > ### Author Response · Authors · 2025-11-30
> > >
> > > As the reviewer acknowledged our clarifications and did not raise further concerns, we hope the AC will take our rebuttal into account when determining the final recommendation.

---

### Official Review · Reviewer_cYyR · 2025-10-31

**Soundness:** 4
**Presentation:** 2
**Contribution:** 3
**Rating:** 6
**Confidence:** 4

**Summary:**

The paper mitigates system prompt extraction (a practical sub-task of direct prompt injection) by encoding the system prompt into soft embeddings. ProxyPrompt replaces the original discrete system prompt with the continuous optimizable embeddings (soft prompts) with the same token length. For a specific system prompt to be protected, the paper optimizes the corresponding soft prompts, aiming at that (1) the new soft prompts function the same as the original system prompt in utility (without any attack); (2) any attempts (user prompts) trying to steal the original prompt will not elicit the model to output it, or a string that functions similarly as the system prompt. Comprehensive experiments show that ProxyPrompt effectively protects various system prompts from five tasks on three models, greatly outperforming the baseline.

**Strengths:**

1. The paper targets an important problem: the system prompt can be an asset of an LLM application, and once it is stolen successfully by one attacker, the commercial competence of the application is degraded.
2. Concurrent to [1] (that does not count against novelty as it goes public after the submission deadline), the paper proposes a new model-based direction to protect the system prompt. In contrast to prompting or guardrails, ProxyPrompt enlarges defender’s optimizable space (multiple soft embeddings) to unlock stronger protection, with understandable improvements.
3. The protection is supported by comprehensive experiments. On five benchmarks, three models, the authors evaluate against 193 attack queries from multiple papers, and compare with three defense baselines. I appreciate the proposed Semantic-Match (SM) and Most-Similar (MS) metrics, which count successfully extracted paraphrased system prompts that function similarly.

**Weaknesses:**

1. Despite the promise of this new defense direction, it has an inherent drawback. If the system developer wants to modify parts of the system prompt, the proxy prompt optimization should be redone. The computational overhead, despite reasonable, hinders the development and iteration of the system. Thanks for reporting the cost in Appendix G (For GSM8K, optimization takes approximately 6 hours with L-70B, 30 minutes with L-8B, and 25 minutes with P-3.8B. For other tasks such as CoLA, the optimization times are 2.5 hours, 18 minutes, and 12 minutes).
2. It is not new to optimize soft prompts for LLM safety/security. Optimizing exactly the same parts (newly added soft prompts), [2] prevents jailbreaks and [3] prevents prompt injections. Even the optimization goal is different, the overall technical novelty is limited, and the authors do not discuss those related work.

3. The presentation is not very clear. Fig 1 is too complex with long texts. Fig 2 illustrates the complex loss design, but I expect to see the defense framework somewhere, see Fig 4 in [2] and Fig 1 in [3]. The presentation of method (using P, Q, P’, Q’) and results (using multiple metrics) are also hard to interpret.

[1] You Can’t Steal Nothing: Mitigating Prompt Leakages in LLMs via System Vectors, CCS’25

[2] On Prompt-Driven Safeguarding for Large Language Models, ICML’24

[3] Defending Against Prompt Injection with a Few DefensiveTokens, AISec’25

**Questions:**

The paper is sound and well-motivated. I don't think my score will likely change by rebuttal interactions, as the mentioned weaknesses are inherent or require naunced revisions. I am looking forward to a new paper version if the authors have time.

---

> ### Author Response · Authors · 2025-11-20
>
> We thank the reviewer for recognizing the importance of protecting system prompts and for acknowledging ProxyPrompt as a novel defense with stronger protection improvements. We also appreciate their positive remarks on our comprehensive evaluation and the proposed leakage detection metrics. We would like to address their concerns below.
>
> > W1. Computational overhead.
>
> We acknowledge that the computational overhead is an important consideration and thank the reviewer for finding it reasonable. We encourage developers to finalize system prompts before applying ProxyPrompt to avoid frequent updates and protect the sensitive parts. As discussed in Line 454, protecting a prompt entirely is often unnecessary because non-sensitive instructions pose no security risk, e.g., “You are ChatGPT, a large language model trained by OpenAI.” We show that proxy prompts remain valid even when non-sensitive instructions are later added, eliminating the need for re-optimization.
>
>
> > W2: Compared with related soft prompt optimization works.
>
> We have discussed relevant soft prompt optimization works in the Related Work section (Prompt design and optimization) and will include the paper [2, 3] in the revised version. While both our work and prior methods involve learning soft prompts, our contributions differ in insight and mechanism:
>
> 1. We leverage the **decoding gap** between soft prompts and tokens to improve prompt security. Prior work considers soft prompts only as a parameter-efficient fine-tuning method against specific attacks like jailbreaks or injection.
> 2. We **replace** system prompts with soft prompts to conceal the original information, whereas prior work **prepends** soft prompts to the input (i.e., soft prompt + system prompt + user query).
>
> Additionally, we introduce semantic-level metrics to more accurately assess prompt leakage. We will revise the paper to clarify our contributions and strengthen its positioning.
>
> > W3: Suggestion on clarity.
>
> We appreciate the suggestion and will revise the paper accordingly to improve clarity in figures and notation.

---

### Note · Authors · 2026-01-02

I have read and agree with the venue's withdrawal policy on behalf of myself and my co-authors.